# Implementing organicity investigations in early psychosis: Spreading expertise

**Jean-Luc Kurukgy** [1], **Julie Bourgin**[2], **Jean-Pierre Benoit**[1], **Sélim Benjamin Guessoum**[3,4,5], **Laelia Benoit**[4,5,6] *

1 Hôpital Delafontaine, Secteur de Psychiatrie Infanto-Juvénile, Saint-Denis, France, 2 Psychiatrie de l'enfant et de l'adolescent–Site Orsay, GH Nord-Essonne, Bures sur Yvette, France, 3 University of Paris, PCPP, Boulogne-Billancourt, France, 4 Child and Adolescent Department—Maison de Solenn, Hospital Cochin, Paris, France, 5 Paris-Saclay University, UVSQ, Inserm 1178, CESP, Team DevPsy, Villejuif, France, 6 Yale School of Medicine (Child Study Center), Yale University, New Haven, CT, United States of America

* laelia.benoit@yale.edu

**Data Availability Statement:** All relevant data are within the paper and its Supporting Information files.

**Funding:** The authors received no specific funding for this work.

## Abstract

### Background

Many medical disorders may contribute to adolescent psychoses. Although guidelines for thorough organicity investigations (OI) exist, their dissemination appears scarce in nonacademic healthcare facilities and some rare disorders remain undiagnosed, many of them presenting without easily recognized phenotypes. This study aims to understand the challenges underlying the implementation of OI in non-academic facilities by practitioners trained in expert centers.

### Methods

Sixteen psychiatrists working at French non-academic facilities were interviewed about their use of OI for adolescents suspected of early psychosis. Interviews were analyzed with Grounded Theory.

### Results

Organicity investigations were found to be useful in rationalizing psychiatric care for the young patient all the while building trust between the doctor and the patient's parents. They also are reassuring for psychiatrists confronted with uncertainty about psychosis onset and the consequences of a psychiatric label. However, they commonly find themselves facing the challenges of implementation alone and thus enter a renunciation pathway: from idealistic missionaries, they become torn between their professional ethics and the non-academic work culture. Ultimately, they abandon the use of OI or delegate it to expert centers.

### Conclusion

Specific hindrances to OI implementation must be addressed.

**Competing interests:** The authors have declared that no competing interests exist.

## Introduction

More than 60 diseases are known to increase the risk of psychotic disorders in childhood and adolescence; they include genetic syndromes, inborn errors of metabolism, and autoimmune, neurologic, endocrinological, and nutrition disorders [1–3]. As many as 12.5% of cases of childhood psychosis may have a medical (somatic) disorder contributing to the clinical presentation (ex: homocystinuria or intermittent porphyria) but their frequency in non-academic settings may be inferior given their more generalist patient recruitment [3]. The involvement of such a disorder should be considered when the following signs are present: visual hallucinations, confusion, catatonia, fluctuating symptoms and intellectual deficiency, as well as when the course appears abnormal (early onset, sudden onset, progressive cognitive decline, or treatment resistance) [4].

Although relevant guidelines for organicity investigations are available [1, 3, 5, 6], many of the disorders present without easily recognized phenotypes and are thus easily missed [7]. Nevertheless, it is important to ensure these diagnoses are made, because even if most of psychoses of organic origins remain inaccessible to specific treatment, some can benefit from suitable treatment and significant clinical improvement [8]. Diagnosing medical and genetic causes and risk factors of early psychosis is a challenge [3]. This challenge includes both training of psychiatrists and implementing organicity investigations throughout all psychiatric units. Eliminating the research-to-practice gap is a challenge of implementation science [9]. Eliminating the training-to-practice gap is another. Implementing changes in medical practice is a major matter that needs research. Collecting the professionals' perspective on such matters is an important aspect of implementation research [10].

Community-based centers, nonacademic hospital departments of child psychiatry, and one-stop youth-friendly medical services offer generalist care for a broad range of disorders [11, 12]. On the contrary, "expert centers" are at the forefront of research on psychosis and provide complete psychiatric assessments, including thorough organicity investigations. Most of them are part of university hospitals.

How can organicity investigations provided in specialized expert centers be implemented effectively in nonacademic facilities? This study explores how practitioners trained in expert centers but working in nonacademic facilities use organicity investigations, including physical assessment and relevant laboratory workups, for their young patients with suspected early psychosis. We believe their personal perspectives will help identify the barriers to organicity investigations' implementation and overcome them.

## Material and methods

This qualitative research was conducted in compliance with the COREQ guidelines [13]. Participants were recruited via the mailing lists of three professional associations of psychiatrists working in the Paris region. The participants had training in organic causes of psychosis at expert centers but were currently working in nonacademic facilities in which they routinely treated adolescents and young adults. One-hour semi-structured face-to-face interviews were conducted by a male fourth-year resident in psychiatry (JLK) and took place in the participants' office with no third-party present. They focused on their daily use of organicity investigations in their workplaces, their training and professional experience, the reactions of their colleagues, patients, and families to these organicity investigations, and their contacts with expert centers. The participants were not made aware of the interviewer's objective and no relationship was established prior to the interview. The interviews were sound recorded, transcribed and analyzed with Grounded Theory [14], a standard methodology for social science research [15], which is particularly suited to understanding professional practices and

organizational cultures. Grounded Theory links social structures with processes occurring at an individual level by focusing on themes that represent underlying interactions and their consequences [16]. The transcripts were not returned to the participants for comment or correction. As in other inductive methods, no exact number of respondents was required before the research began. The data were coded to generate categories, which were then validated through constant comparisons as new interviews were performed [17]. Thus, data analysis, further sampling, and theoretical development proceeded simultaneously until saturation was reached [14]. To ensure reliability, two researchers (JLK, LB (female; PhD) independently coded and analyzed all data and their findings were discussed during research team meetings, a process called triangulation [18]. No software was used during this process to manage the data. Consistency between the data and the findings was assessed by triangulation and participant feedback. No ethical approval was necessary since no patient data was used in this study. All participants gave informed consent.

## Results

Sixteen psychiatrists (4 male and 12 female) aged 28 to 60 years (mean: 37; SD: 9,38) were selected through snowball sampling and all of them agreed to participate in the present study (see Table 1). Four were residents, six were fellows and seven were senior practitioners. Most participants (14/16) had subspecialized in child and adolescent psychiatry during their medical studies. Four participants provided direct feedback to the researchers and agreed with the findings. The results of the qualitative analysis are summarized in Fig 1. In this figure, major themes are presented with a blue background and minor themes with a withe background. For further details on the analysis, see S1 Table. Organicity investigations were found to be useful in rationalizing psychiatric care for the young patient all the while building trust between the doctor and the patient's parents. However, in their quest for implementation, recently trained psychiatrist faced challenges that left neither their practice nor their new workplace unchanged.

### A reassuring relational tool but a short-term avoidance of psychiatry

**Advantages of organicity investigations.** Organicity investigations were found to be a relational tool for families and a reassuring tool for doctors.

Psychiatrists viewed organicity investigations as a way to initiate the medical relationship with the patient's parents whom they expected to be anxious about their first contact with psychiatry. They announced the carrying out of organicity investigations at the earliest opportunity to reassure them and to bespeak their expertise to the patient's family. Besides, organicity investigations postponed facing the possibility of a psychiatric label and its social stigma, a possibility the doctors did not expect families to welcome.

When dealing with adolescents with early and unsettled symptoms, the concrete and scientific nature of organicity investigations helped in managing the medical uncertainty of *"unclear, imprecise, invisible"* psychiatry by providing a feeling of reassuring certainty of the absence of a somatic disorder. Once completed, they provided reliable results and legitimized a shift away from diagnosis and a focus on psychiatric care. Even though, as the saying goes, the absence of evidence isn't the evidence of absence (these investigations only concern the somatic causes that we currently know about), they were therefore seen as a mandatory prerequisite to starting psychiatric care.

**Limitations of organicity investigations.** Organicity investigations are a pretext for short-term avoidance of psychiatry. Furthermore, they are unwelcomed for the patient and constraining for physicians.

**Table 1. Participants.**

| Interview number | Sex | Age | Status | Type of nonacademic setting: Paris region | Previous Training in OI | Has an EC contact | Refers to ECs | Established collaboration |
|---|---|---|---|---|---|---|---|---|
| 1 | F | 28 | Resident | Public youth psychiatric department | Residency in an EC | Yes | No | No |
| 2 | M | 30 | Fellow | Public youth psychiatric department in a general hospital | Residency in an EC | Yes | No | No |
| 3 | F | 48 | Senior practitioner | Public youth psychiatric consultation center | Fellowship in an EC | Yes | For complex diagnoses | Yes |
| 4 | F | 55 | Senior practitioner | Public psychiatric department of psychiatric hospital | Academic courses | Yes | No | No |
| 5 | F | 30 | Resident | Public psychiatric department of psychiatric hospital | Residency in an EC / Academic courses | Yes | No | No |
| 6 | M | 60 | Senior practitioner | Public psychiatric department of a medical and educational center | Residency in an EC / Academic courses | Yes | For complex diagnoses | No |
| 7 | M | 37 | Senior practitioner | Public psychiatric consultation center | Academic courses | Yes | Systematically | No |
| 8 | M | 35 | Fellow | Public youth psychiatric consultation center | Residency in an EC / Academic courses | Yes | For complex diagnoses | No |
| 9 | F | 31 | Fellow | Public psychiatric consultation center | Residency in an EC | Yes | No | No |
| 10 | F | 30 | Fellow | Public youth psychiatric consultation center of general hospital | Academic courses | Yes | No | No |
| 11 | F | 35 | Senior practitioner | Public youth psychiatry consultation center | Residency in an EC / Academic courses | Yes | For complex diagnoses | No |
| 12 | F | 40 | Senior practitioner | Public youth psychiatric department of psychiatric hospital | Academic courses | No | For complex diagnoses | No |
| 13 | F | 35 | Fellow | Public psychiatric department of psychiatric hospital | Academic courses | Yes | Systematically | No |
| 14 | F | 35 | Fellow | Public youth psychiatric consultation center | Fellowship in an EC | Yes | No | No |
| 15 | F | 30 | Resident | Public youth psychiatric department of general hospital | Personal acquaintance with a peer youth psychiatrist trained in an EC | No | No | No |
| 16 | F | 34 | Senior practitioner | Public youth psychiatric department of general hospital | Residency in an EC | No | For complex diagnoses | No |

The initial relief provided by the organicity investigations was short-lived, however, because in the end, parents mostly found their child diagnosed with an emerging mental illness. Inversely, the young patients were described as either indifferent or opposed to organicity investigations. Psychiatrists attributed this opposition to psychiatric symptoms (such as delusions) rather than any reasoned decision. There are therefore two relational drawbacks identified by the participants. The first was the fact that the avoidance of dealing with families' worries about their child's long-term mental health, future schooling and social integration was only slightly delayed. Secondly, they sometimes found that trying to convince their patient at any cost generated tension.

Lastly, organicity investigations were seen as constraining, given that they require clinical expertise to identify simultaneously atypical mental symptoms and signs of physical pathology. Participants found the identification of rare diseases a difficult body of knowledge to master and maintain; the skills they had acquired during their training in expert centers gradually

| **The daily effects of organicity investigations** | **Advantages of organicity investigations** | | |
|---|---|---|---|
| | A relational tool for families | A technical tool for doctors | |
| | **Limitations of organicity investigations** | | |
| | Short-term avoidance of psychiatry | Unwelcomed tests for the patients | A constraint for the psychiatrist |

| **To transform:**<br><br>**Implementing organicity investigations as a recently trained psychiatrist** | **Young experts joining an aging institution** | | |
|---|---|---|---|
| | A sense of duty | A driving force for implementation | Disseminating knowledge tactfully |
| | **Reshaping the use of organicity investigations** | | |
| | Perseverance | Renunciation | Pragmatic patchwork |
| | **Seeking external support** | | |
| | Referral | Partnership | Accurate somatic assessments | Experiencing loneliness |

| **To be transformed:**<br><br>**Experience changes practices** | **The resident on a mission** |
|---|---|
| | Professional ethics: a central driving force |
| | **The rise of an inner conflict** |
| | Protocols and compromises to adjust to organizational constraints |
| | **Senior practitioners who give up** |
| | Disavowal of their own ideals and betrayal of their teachers |

**Fig 1. Main results—the challenges of organicity investigations implementation in psychosis.**

fading. Furthermore, discouraging technical obstacles turned organicity investigations a meaningless and time-consuming procedure for psychiatrists working in nonacademic facilities.

## To transform: Implementing organicity investigations as a recently trained psychiatrist

**Young experts joining an aging institution.** Young experts joining an aging institution feel the duty to implement medical progress within the psychiatry department and must disseminate the knowledge tactfully.

Recently trained participants considered themselves ethically bound to prescribe up-to-date medical investigations, such as organic and neurometabolic workups. The importance of organicity investigations seemed rooted in their belief that such investigations preserved their identity as medical doctors.

Given their background in expert centers and their senior colleagues' lack of knowledge about organicity investigations, residents found themselves in the position of experts in their new workplace and considered it a duty to be a driving force for their implementation. Some described a messianic view of their role in transmitting expert center know-how. However, the specificity of organicity investigations, on the edge of medicine and psychiatry, sometimes generated disinterest or rejection which encouraged some participants to tread lightly and disseminate their knowledge tactfully. Attempts to implement organicity investigations, despite colleagues' *"somewhat hermetic attitude"* sometimes resulted in a segmentation of care or a delegation to practitioners of other specialties or expert centers, and sometimes aggravated political tensions between hospital departments, with the refusal to change practices resulting from institutional disagreements.

**Reshaping the use of organicity investigations.** Faced with the constraints of their new workplace, participants either persevered in their use of organicity investigations, renounced, or settled for a pragmatic patchwork of tests.

Training in organicity investigations, however complete, did not appear to guarantee that the participants performed them in their nonacademic workplaces. None of the psychiatrists disputed the importance of organicity investigations in addressing emerging psychosis in adolescents. The constraints of their daily work reshaped their use, however:

The practices of some participants remained modeled on that of the expert center they had been trained at. They inflexibly ordered the same set of examinations, thus conforming to the primacy of evidence-based medicine over the work organization of their non-academic facility. Other participants explained that the lack of support from their colleagues had exhausted their initial enthusiasm and led them to gradually halt organicity investigations orders within a few months. Meanwhile, the rarity of conclusive evidence in daily practice confirmed their abandonment without systematically referring patients to an expert center. Lastly, some adopted a more pragmatic method by taking into account the financial constraints of nonacademic facilities and prioritizing less costly explorations. Similarly, local technical possibilities sometimes dictated which investigations were done. Although only partially satisfactory, such strategies allowed a pragmatic compromise between medical requirements and the reality on the ground.

**Seeking external support.** Our participants feel they need external support in order to refer patients, partner up for complex cases and get accurate somatic assessment. Those without a professional network experience loneliness.

The specific hindrances arising from the use of organicity investigations in non-academic workplaces and the reshaping it necessitates lead participants in a search for support and

collaboration. For nonacademic psychiatrists, sharing common challenges with expert center colleagues enabled:

- a collegial decision-making process about complex cases, therefore splitting the heavy medical responsibility of diagnosis and treatment choice.

- mediation of the relationship with the patient and his or her family by providing external and expert validation of the proposed care, thereby strengthening the therapeutic alliance.

Despite their training, some participants perceived former expert center colleagues as more competent and consulted them in order to refer their patients. Nonetheless, due the prestige of expert centers, their relations with their former colleagues sometimes generated feelings of misunderstanding, harsh judgments, and lack of dialogue between these two distinct worlds.

Furthermore, the rarity and the diversity of diseases sought require a dialogue between psychiatrists and somatic specialists for accurate somatic assessment and optimal care. However, such cooperation was found to be difficult to establish. The psychiatrists' enthusiasm for organicity investigations contrasted with the lack of commitment of the general practitioners working in nonacademic facilities. Therefore, participants were worried about the stigmatization of psychiatric patients would lead to less conscientious assessments.

Practitioners who did not have a wide professional network experienced difficulty in finding specialists with whom to share their uncertainties. Consequently, the autonomy granted to the trained psychiatrists gave them the freedom to practice as they saw fit but generated loneliness and isolation.

## To be transformed: Changes in practices, as psychiatrists gain experience

The use of organicity investigations evolves across the recently trained psychiatrists' working lives. Thus, we can sketch a typical career path that often leads to the abandonment of the practices recommended by the expert center they trained in because of the constraints of their work in nonacademic departments. Each of these stages will be described by the ideal type defined by Weber, the aim of which is to highlight the most significant traits to enable a better understanding of the social action involved [19].

**The resident on a mission.** Young residents, driven by their professional ethics, are keen to spread the use of organicity investigations.

Convinced by their teachers, young practitioners want to apply their expertise in their new environment. They feel they have a mission: to save their young patients long years of psychiatric care by diagnosing a curable disease. Accordingly, they are usually keen to spread this practice, they do not hesitate to manage complex clinical situations, or use their network in expert centers to seek help. In nonacademic settings where no protocol covers organicity investigations, professional ethics drive its pursuit.

**The rise of an inner conflict.** Over time, professional duties and local constraints generate doubt regarding the use of organicity investigations.

An intensive work rhythm puts daily pressure on the initial enthusiasm of these recently trained practitioners. They come to doubt the possibility of pursuing the use of organicity investigations as they learned it while fulfilling their responsibilities in their nonacademic setting. Aware of the inconsistency between the time-consuming use of organicity investigations and their department's inaction in regard to organizational planning, they begin to advocate a systematization of practice by developing departmental protocols. A compromise solution is then adopted to apply organicity investigations routinely: they no longer try to clinically identify atypical symptoms that justify carrying out targeted complementary examinations, but rather automate the use of organicity investigations to facilitate it. These explorations are then

chosen based on financial constraints and technical possibilities rather than on clinical relevance. Thus, prescribing organicity investigations turns into a meaningless routine.

**Senior psychiatrists who give up organicity investigations.** Ultimately, trained psychiatrists renounce the use of organicity investigations, but trust expert centers will go on.

Senior participants admitted that they were overwhelmed and *"saddened"* by the fact that their efforts at change had failed to move their teams out of the *'listlessness"* of nonacademic departments. They now referred their patients to expert centers. Their awareness of their renunciation resonated as a painful disavowal of their former commitments and as a betrayal of their teachers.

## Discussion

In this study, the obstacles to the implementation of organicity investigations came from the institution's routine and overall inertia (leaving aside budgetary and technical problems). Psychiatrists and patients' relatives find organicity investigations reassuring at a time when they are faced with the possibility of psychosis onset. In contrast, convincing the patient sometimes proved difficult, and psychiatrists tried to provide guidance and balanced information during these trying times [20]. Their goal was to avoid the entanglement of care and coercion which affects caregivers, the patients and their families [21, 22]. Their feeling of exigency to implement organicity investigations may also originate from the relatively high frequency of psychoses of organic origin in the expert centers they trained at. Nonetheless, the generalist care nonacademic facilities are assigned to perform, the high level of expertise required for organicity investigations, and the rarity of the diseases investigated ultimately led to unsatisfactory use of organicity investigations or systematic referral of patients to expert centers. The participants also felt isolated from the institutions they had trained in and ultimately lost contact with colleagues to consult when needed.

The science of implementation provides a relevant framework for the analysis of these findings and may help explain the gap between evidence-based practices and what is provided to consumers in routine care [9, 23–27]. Indeed, obstacles to change practices can arise at several levels in the healthcare system: the patient, the professional, team, organization, or environment. In the field of mental health, 15 to 20 years can separate the establishment of such practices and their widespread generalization, a gap that prevents patients from reaping the benefits of costly research [27].

### Relational advantages of organicity investigations

Our results showed that organicity investigations were not only used as a technical medical tool but were especially considered as a way to manage the relationship and build trust with patients and their families. Indeed, despite the fact that these disorders are also associated with functional remission [28, 29], psychiatrists envision psychosis as a disastrous life-long disorder [30] and this viewpoint shapes their reluctance to disclose psychosis risks, a choice related to their belief in the self-fulfilling prophecy [31], and their knowledge of the stigma generated by a psychiatric label [32–35]. Organicity investigations thus convey a reassuring message: it may be possible to avoid psychiatric diagnosis and find a curable illness instead.

The prospect of early psychosis in a patient exposes practitioners to the uneasiness of medical uncertainty. Indeed, doctors face many uncertainties inherent in medical knowledge and are thus trained to remain in control in their daily practice [36, 37]. Psychiatry is especially subject to uncertainty because of its current technical and theoretical immaturity. The proper use of organicity investigations at the onset of psychosis is a good example: recommendations are fairly recent, not widely disseminated, and research is still embryonic. In completing

academic courses in organicity investigations or training in expert centers, participants attempted to control medical uncertainty and as the same time used all of the latest medical tools available for their young patients' care. They considered, as do we, that no distinction should be made between psychiatry and "somatic medicine". Thus, organicity investigations are an integral part of a psychiatrist's job.

## Institutional obstacles for organicity investigations implementation

Our participants have experienced the difficulties of implementing alone a novel medical practice in a workplace with its own work culture and habits. Laypeople may think that medical knowledge and techniques are widely and rapidly adopted by the medical community when their superiority is established. This process is not nearly so straightforward, however. Behind the indisputable assertions of "finished science" lie many controversies and decisions not solely related to science. Studying science "in the making" makes it possible to pinpoint the moment when scientific discoveries might have gone in many other directions [38]. Indeed, this study depicts the intricacies of the adoption of organicity investigations: rivalries between departments and professionals, budgetary and technical criteria, as well as institutions' desire (or lack thereof) for change.

This highlights the existence of segments in the medical profession [39]. Behind its apparent homogeneity lie many disparities—in values, interests, and methodologies—that segment it, creating opposing dynamic forces that can lead to institutional change. They also produce an informal division of labor, with each segment delegating work to another [40–42]. In our study, the segment of young experts urges the opposing segment—their nonacademic colleagues—to implement organicity investigations, with various outcomes. The multiplicity of parties involved in the decision-making process (psychiatrists, other specialists, administrators) and its centralized nature make any change long and its outcome uncertain. This "bureaucratic phenomenon", characterized by its rigidity and inefficiency [43], becomes a source of tedious work (time-consuming paperwork and procedures) and frustration which clashes with participant's professional ethics.

Furthermore, various characteristics of research evidence affect its usability in clinical practice. For instance, change is difficult if the innovation requires complex changes (in the decision-making process or the acquisition of new skills), better collaboration between disciplines, or changes in the organization of care. Adherence to new guidelines also depends on the type of health issue: it is better for acute than chronic care. All of these findings help explain the difficulty the participants have had in implementing organicity investigations in their nonacademic facilities: it concerns a chronic health issue and necessitates complex clinical evaluations and close collaboration between practitioners of different specialties. Targeted interventions (small group interactive training, feedback on performance, computer assistance), which have been shown to be the most effective way to transfer evidence into practice, might usefully be applied to improve the dissemination of organicity investigations [26].

## Interactional obstacles for organicity investigations implementation

The newly trained participants are the only ones with strategic knowledge on organicity investigations, they are the necessary link between two systems that otherwise have trouble communicating: the expert center and the nonacademic department. It has however been shown that better collaboration and recognizing the "felt" power differential between institutions are essential to improve relations in trying times such as psychosis onset [44]. The newly trained participants therefore are at the center of a creative process that can lead to the adoption of organicity investigations. It is nonetheless a challenging position because they are not

recognized as an integral part of either system, and they risk isolation when difficulties arise [45]. Interactions between the participants and their colleagues led to what A. Strauss calls "self-appraisals" and ultimately to the adoption of new ethics, or alternatively to the abandonment of old ones [46]. Furthermore, the lack of support from the organization's leadership– experienced by some participants–is known to be a major barrier to implementation of new guidelines in routine practice [25].

According to Becker, medical trainees never lose their idealistic view of medicine; they simply adapt realistically to the situations they face during their training. By interacting with their peers and teachers, they become "pragmatically idealistic" [47]. We have shown this transformation by defining a "career path" in three characteristic stages, focusing not on *who* the participants are, but on *what* they are doing, and thus describing the social processes underlying their behavior [48, 49]. Indeed, daily interactions come into play over the years and lead young practitioners away from their mission by making them doubt the possibility of implementing organicity investigations in their nonacademic department. Ultimately, these processes can lead them to a heavyhearted abandonment or delegation of its use.

## Conclusion

Implementing organicity investigations in nonacademic departments is not simply a matter of access to training in expert centers but necessitates facing relational, institutional and interactional challenges which involve facing the medical uncertainty inherent to psychosis onset and an extensive reorganization of care. Surprisingly, we also found organicity investigations to be a tool to build trust between physicians and the patients' parents. Further research might be helpful in quantifying the current benefit for patient care in regard to the time and effort conceded by professionals. Tackling these challenges is nonetheless essential to guarantee the best care available to young patients, especially because expert centers cannot shoulder the burden of these investigations alone. To that end, we suggest the following measures:

---

**Proposed initiatives to facilitate organicity investigations' implementation in nonacademic departments:**

- Establish **consensual international official guidelines** for thorough medical assessment when childhood schizophrenia is suspected and enforce **regular inspections of practices**.
- Develop **targeted interventions** (small group interactive training, feedback on performance, computer assistance) in nonacademic departments.
- Develop **the use of shared positions** between nonacademic departments and expert centers for psychiatrists to ensure a long-lasting relationship between practitioners.
- Assign the "shared" psychiatrist mentioned above to be responsible for **making referrals for complex cases to expert centers.**
- Further develop **joint research programs** which could also improve dialogue between expert centers and nonacademic facilities.
- Sustained collaboration and training with somaticians (such as pediatricians, neurologists. . .) may improve the psychiatrists' approach and knowledge of somatic diseases and strengthen the partnership in patient care

## Supporting information

**S1 Table. Analysis details.**
(DOCX)

## Author Contributions

**Conceptualization:** Jean-Luc Kurukgy, Julie Bourgin, Jean-Pierre Benoit, Laelia Benoit.

**Data curation:** Jean-Luc Kurukgy.

**Formal analysis:** Jean-Luc Kurukgy, Laelia Benoit.

**Investigation:** Jean-Luc Kurukgy.

**Methodology:** Julie Bourgin, Jean-Pierre Benoit, Laelia Benoit.

**Supervision:** Julie Bourgin, Jean-Pierre Benoit, Laelia Benoit.

**Writing – original draft:** Jean-Luc Kurukgy, Sélim Benjamin Guessoum.

**Writing – review & editing:** Jean-Luc Kurukgy, Sélim Benjamin Guessoum, Laelia Benoit.

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
