## [Decision Letter · Decision Letter 0]

22 Mar 2021

PONE-D-20-37522

Implementing organicity investigations in early psychosis: Spreading expertise

PLOS ONE

Dear Dr.Kurukgy 

Thank you for submitting your manuscript to PLOS ONE. After careful consideration, we feel that it has merit but does not fully meet PLOS ONE’s publication criteria as it currently stands. Therefore, we invite you to submit a revised version of the manuscript that addresses the points raised during the review process.

One of the major areas of that must be addressed is the description, meaning  and perhaps renaming of 'youth psychiatrists' Kindly address all of the comments made by all of the reviewers. Please submit your revised manuscript by April 17th, 2021.  If you will need more time than this to complete your revisions, please reply to this message or contact the journal office at plosone@plos.org. Please include the following items when submitting your revised manuscript:

We look forward to receiving your revised manuscript.

Kind regards,

Gerard Hutchinson, MD

Academic Editor

PLOS ONE

Journal Requirements:

b) If there are no restrictions, please upload the minimal anonymized data set necessary to replicate your study findings as either Supporting Information files or to a stable, public repository and provide us with the relevant URLs, DOIs, or accession numbers. Please see http://www.bmj.com/content/340/bmj.c181.long for guidelines on how to de-identify and prepare clinical data for publication. For a list of acceptable repositories, please see http://journals.plos.org/plosone/s/data-availability#loc-recommended-repositories

4. Please upload a copy of Figure 1, to which you refer in your text. If the figure is no longer to be included as part of the submission please remove all reference to it within the text.

Additional Editor Comments (if provided):

Reviewers' comments:

Reviewer's Responses to Questions

**Comments to the Author**

1. Is the manuscript technically sound, and do the data support the conclusions?

Reviewer #1: Yes

Reviewer #2: Partly

Reviewer #3: Yes

2. Has the statistical analysis been performed appropriately and rigorously? 

Reviewer #1: N/A

Reviewer #2: N/A

Reviewer #3: Yes

3. Have the authors made all data underlying the findings in their manuscript fully available?

Reviewer #1: Yes

Reviewer #2: Yes

Reviewer #3: Yes

4. Is the manuscript presented in an intelligible fashion and written in standard English?

Reviewer #1: Yes

Reviewer #2: Yes

Reviewer #3: Yes

5. Review Comments to the Author

Reviewer #1: The authors were interested in a subject that has not yet been widely treated: the practical realization of organicity investigations (OI), particularly barriers and drivers to them, in adolescents with early psychoses treated in non-academic psychiatric centers, by interviewing French psychiatrists trained in expert centers about their practices.

The authors identified several advantages to this OI: facilitating the initiation of a trusting relationship (especially with the patients' relatives), having a "technical" tool to hold on to, corresponding to a certain professional identity (notably professional requirement and conscience), and disadvantages: complexity of the practical realization, negativity in the vast majority of cases leading to a psychiatric diagnosis of difficult acceptance, high level of competence required, difficulties in changing practices and a consequent feeling of isolation The authors pointed out the evolution of practices between the training received in expert centers ("the ideal") and the daily life then implemented by the practitioners ("the reality"), with a certain diversity of the professionals' paths (compromise between these 2 visions, abandonment...).

This research is very interesting because the subject is crucial given the potentially curable nature of certain diseases. The authors underline the extreme complexity of the subject given the absence of international consensual guidelines, the need for important competences and close links with paediatricians and expert centers given the large number of pathologies to be evoked, links that are not always well protocolized in practice. The discussion is well structured, and the authors propose measures to improve the situation as an opening remark.

There are no major problems in this very well-written and enjoyable article.

Some minor points:

- INTRODUCTION: Nuance the importance of recognizing psychoses of organic origin, most of which unfortunately remain inaccessible to specific treatment.

This point could also be addressed in the discussion, since there seems to be a discrepancy between the perception of the frequency of organically-induced psychoses accessible to treatment (and which should therefore not be "missed") and the statistical reality of these very rare situations, a discrepancy that may be accentuated by the expert centers, which themselves only (or almost only) see organically-induced psychoses. This is perhaps partly responsible for the feeling of exigency of the newly trained practitioners, and may give rise to a "messianic" vision of their work in non-academic centers.

- RESULTS. 1.1 Caution about the impression of reassurance induced by the "elimination" of any somatic cause. "Absence of evidence is not evidence of absence".

- DISCUSSION: Clinical practice guidelines: consider a specific point concerning the improvement of collaboration with and training of potentially involved somaticians (pediatricians, general practitioners, neurologists, internists, etc.)

Reviewer #2: This is an interesting study that aimed to understand the challenges underlying the implementation of organicity investigations (OI) in non-academic facilities by practitioners trained in expert centers. I have the following comments:

It is stated that providing medical (somatic) disorders can lead to suitable treatment and significant clinical improvement [for psychosis]. What is the evidence for this? Considering how strong this premiss is, it is important to review the evidence in support for this claim.

It is suggested that organicity investigations in relation to psychosis diagnosis are already provided in specialized expert centres. It is important critically review how much these organicity investigations improve psychosis diagnosis and subsequent treatment outcomes in patients with psychoses.

It is equally important describe how much extract time and effort organicity investigations will require from both clinicians and patients. It they entail more additional time and effort for almost no improvement in diagnosis precision and treatment outcomes in patients with psychosis then they need for implementation non-academic facilities may be questioned.

When describing methods, it is stated that participants recruited in this study had training in organic causes of psychosis at expert centers but were currently working in non-academic facilities. Do the authors mean psychiatrists, trained psychologists or students? Please provide more detail on such an important aspect of the study.

The description of the sample in the results section is also necessary including mean and SD of age, gender distribution years of experience and level of educational attainment.

Also please explain what “youth psychiatrists” are. I assumed it was referring to psychiatrists who were young, but this may be incorrect interpretation.

It is quite misleading and factually inaccurate to say that psychosis as a disastrous life-long disorder. Over 60% of people with a diagnosis of psychosis fully recover within the first 2-5 years

The overall purpose of the study is not very clear. From the introduction it sounded like the aims of this work were to investigate how OI can be implemented in non-academic institutions. However, it appears that the actual aims were to identify personal perspectives of people with OI training on the benefits of OI and obstacles of their implementations. This disparity is rather confusing.

The conclusions in this work are based 0n personal opinions of 16 youth psychiatrists, who may be rather naïve or inexperienced to make rational recommendations. To make the conclusions stronger psychiatrists with more years of experiences need to be included in this study. Additionally, evidence from research studies, such as CRT, in support to the conclusions would be helpful.

Reviewer #3: Very useful article!

Althoug some wording revision should be applied:

page 3, last sentence. replace semicolons by commas.

Page 4: first sentence: Sixteen "youth" psychiatrists aged 28 to 60. I suggest to remove the word "youth" given the fact that the age range goes far beyong youth.

Page 7: from third line: very long sentence. Try to improve wording by separating sentences.

Page 8: "The practices of some participants remained modeled on that of the expert center they had [been]

trained at." The verb on the sentence was missing.

Page 9: wording suggestions: "Therefore, participants [were]worried [about] the

stigmatization of psychiatric patients would lead to less conscientious assessments"

Figure 1: I don't fully grasp the difference between the second ("to transform") and the third ("to be transformed") categories. I suggest to choose other words to highlight the core ideas

6. PLOS authors have the option to publish the peer review history of their article (what does this mean?). If published, this will include your full peer review and any attached files.

Reviewer #1: No

Reviewer #2: No

Reviewer #3: **Yes: **Paula Herrera-Gomez

---

## [Author Response · Author response to Decision Letter 0]

20 Apr 2021

Response to reviewers – Implementing organicity investigations in early psychosis: Spreading expertise

Dear editorial team, Dear reviewers, 

It is with great pleasure that we submit for your consideration our revised manuscript. After careful examination of each point raised by the academic editor and the reviewers, the necessary changes have been made and are highlighted in the file, as requested. 

The term “youth psychiatrist”, used in our article to designate psychiatrists specialized in adolescents’ and young adults’ mental health, has been found to be confusing and unclear. This issue has been addressed by further detailing the participants’ status and specialty as well as their work settings. 

Following is a detailed list of the changes made to comply to the journal requirements:

1. The manuscript now meets PLOS ONE's style requirements, including those for file naming, as detailed in the formatting guidelines.

2. There are no ethical or legal restrictions on sharing de-identified data. Therefore, you will find in “S1_File” the minimal anonymized data set necessary to replicate the study findings

3. The “ethics” statement has been moved to the required location and the “competing interest” statement has been removed from the manuscript

4. Figure 1 has been uploaded in the manuscript. It can be found at the beginning of the Result section

5. Captions for the Supporting Information file has been included at the end of the manuscript and the in-text citation has been updated to match accordingly

Response to Reviewer 1 : 

• INTRODUCTION: Nuance the importance of recognizing psychoses of organic origin, most of which unfortunately remain inaccessible to specific treatment.

This point could also be addressed in the discussion, since there seems to be a discrepancy between the perception of the frequency of organically-induced psychoses accessible to treatment (and which should therefore not be "missed") and the statistical reality of these very rare situations, a discrepancy that may be accentuated by the expert centers, which themselves only (or almost only) see organically-induced psychoses. This is perhaps partly responsible for the feeling of exigency of the newly trained practitioners, and may give rise to a "messianic" vision of their work in non-academic centers.

• We agree with the reviewer and have modified the manuscript as follows: 

“As many as 12.5% of cases of childhood psychosis may have a medical (somatic) disorder contributing to the clinical presentation (ex: homocystinuria or intermittent porphyria) but their frequency in non-academic settings may be inferior given their more generalist patient recruitment (1).” And “Nevertheless, it is important to ensure these diagnoses are made, because even if most of psychoses of organic origins remain inaccessible to specific treatment, some can benefit from suitable treatment and significant clinical improvement (2).” And in the Discussion section: “Their messianic enthusiasm to implement organicity investigations may also originate from the relatively high frequency of psychoses of organic origin in the expert centers they trained at.”

• RESULTS. 1.1 Caution about the impression of reassurance induced by the "elimination" of any somatic cause. "Absence of evidence is not evidence of absence".

• We thank the reviewer for this comment and for providing us this adage which highlights the subjective nature of the reassuring feeling provided by organicity investigations. We have taken the liberty to insert it as is in the manuscript: 

“(…) the concrete and scientific nature of organicity investigations helped in managing the medical uncertainty of “unclear, imprecise, invisible” psychiatry by providing a feeling of reassuring certainty of the absence of a somatic disorder. Once completed, they provided reliable results and legitimized a shift away from diagnosis and a focus on psychiatric care. Even though, as the saying goes, the absence of evidence isn’t the evidence of absence (these investigations only concern the somatic causes that we currently know about), they were therefore seen as a mandatory prerequisite to starting psychiatric care.”

• DISCUSSION: Clinical practice guidelines: consider a specific point concerning the improvement of collaboration with and training of potentially involved somaticians (pediatricians, general practitioners, neurologists, internists, etc.)

• We agree with the reviewer and have added a specific point in the manuscript: 

“Sustained collaboration and training with somaticians (such as pediatricians, neurologists…) may improve the psychiatrists’ approach and knowledge of somatic diseases and strengthen the partnership in patient care”. Following the comment of Reviewer 2, the “Clinical Practice Guidelines” have been retitled “Proposed initiatives to facilitate organicity investigations’ implementation in nonacademic departments”.

Response to Reviewer 2 :

• It is stated that providing medical (somatic) disorders can lead to suitable treatment and significant clinical improvement [for psychosis]. What is the evidence for this? Considering how strong this premiss is, it is important to review the evidence in support for this claim.

• We agree with the reviewer and have added a citation to back the statement that suitable treatment can lead to clinical improvement for psychosis of organic origins. (Merritt J, Tanguturi Y, Fuchs C, Cundiff AW. Medical Etiologies of Secondary Psychosis in Children and Adolescents. Child and Adolescent Psychiatric Clinics of North America. 2020 Jan 1;29(1):29–42).

• It is suggested that organicity investigations in relation to psychosis diagnosis are already provided in specialized expert centres. It is important critically review how much these organicity investigations improve psychosis diagnosis and subsequent treatment outcomes in patients with psychoses.

• We agree with the reviewer and have specified which citation illustrates the impact of organicity investigations on psychosis diagnosis and treatment (Giannitelli, M., Consoli, A., Raffin, M., Jardri, R., Levinson, D. F., Cohen, D., & Laurent-Levinson, C. (2018). An overview of medical risk factors for childhood psychosis: Implications for research and treatment. Schizophrenia Research, 192, 39–49. https://doi.org/10.1016/j.schres.2017.05.011)

• It is equally important describe how much extract time and effort organicity investigations will require from both clinicians and patients. It they entail more additional time and effort for almost no improvement in diagnosis precision and treatment outcomes in patients with psychosis then they need for implementation non-academic facilities may be questioned.

• We agree with the reviewer that the improvement in diagnosis and treatment has yet to be compared to the extra time and effort put into their carrying out in nonacademic settings. We have not found studies addressing this topic and this was one of our main motivation for conducting this research. However, our data suggests that the effort and time needed is currently perceived as a significant obstacle to sustained use of organicity investigations (cf. “Reshaping the use of organicity investigations” in the Results section). This limitation was highlighted in the Conclusion: “Further research might be helpful in quantifying the current benefit for patient care in regards to the time and effort conceded by professionals.”

• When describing methods, it is stated that participants recruited in this study had training in organic causes of psychosis at expert centers but were currently working in non-academic facilities. Do the authors mean psychiatrists, trained psychologists or students? Please provide more detail on such an important aspect of the study.

The description of the sample in the results section is also necessary including mean and SD of age, gender distribution years of experience and level of educational attainment.

• We thank the reviewer for this comment and have provided the required information: “Sixteen youth psychiatrists (4 male and 12 female) aged 28 to 60 years (mean: 37 ; SD : 9,38) and working in a variety of settings, were selected through snowball sampling and all of them agreed to participate in the present study (see Table 1). Four were residents, six were fellows and seven were senior practitioners.” 

The type of education attainment regarding organicity investigations is listed for each participant in Fig. 1: “Residency in EC”, “Fellowship in EC” and “Academic courses”. 

• Also please explain what “youth psychiatrists” are. I assumed it was referring to psychiatrists who were young, but this may be incorrect interpretation.

• We thank the reviewer for giving us the opportunity to clarify this point: the term “youth psychiatrist” has been removed as it seemed to be confusing. We have addressed this issue by specifying the participants’ work setting as well as their professional status and level of specialization: “The participants had training in organic causes of psychosis at expert centers but were currently working in nonacademic facilities in which they routinely treated adolescents and young adults. (…) The majority of participants (14/16) had subspecialized in child and adolescent psychiatry during their medical studies. »

• It is quite misleading and factually inaccurate to say that psychosis as a disastrous life-long disorder. Over 60% of people with a diagnosis of psychosis fully recover within the first 2-5 years

• We agree with the reviewer regarding the recovery of people with a diagnosis of psychosis. The identification of psychosis as a “life-long disaster” by psychiatrists originates from a qualitative study addressing the psychiatrists’ representation and attitude towards psychosis risk management 

To highlight the discrepancy between these representations and the reality of the evolution of such disorders, we have modified the manuscript as follows : “Indeed, despite the fact that these disorders are also associated with functional remission (25,26), psychiatrists envision psychosis as a disastrous life-long disorder (27) and this viewpoint shapes their reluctance to disclose psychosis risks, a choice related to their belief in the self-fulfilling prophecy (28), and their knowledge of the stigma generated by a psychiatric label (29–32). »

• The overall purpose of the study is not very clear. From the introduction it sounded like the aims of this work were to investigate how OI can be implemented in non-academic institutions. However, it appears that the actual aims were to identify personal perspectives of people with OI training on the benefits of OI and obstacles of their implementations. This disparity is rather confusing.

• We thank the reviewer for the opportunity to clarify the aim of our study. The overall aim of this study was to enquire to the professionals “on the field” about their usage of organicity investigations to determine the barriers to their implementation. In the framework of implementation science, the evidence-based practice is distinguished from its implementation process and the professionals’ perspectives play an important role. (Bauer, M.S., Damschroder, L., Hagedorn, H. et al. An introduction to implementation science for the non-specialist. BMC Psychol 3, 32 (2015). https://doi.org/10.1186/s40359-015-0089-9). To clarify the interconnexion between the professionals’ perspective and the implementation of organicity investigations, we have modified the Manuscript as follows : “Implementing changes in medical practice is a major matter that needs research. Collecting the professionals’ perspective on such matters is an important aspect of implementation research (10). (…) This study explores how practitioners trained in expert centers but working in nonacademic facilities use organicity investigations, including physical assessment and relevant laboratory workups, for their young patients with suspected early psychosis. We believe their personal perspectives will help identify the barriers to organicity investigations’ implementation and eliminate them.”

To further highlight this, we have also changed the title of our table in the Conclusion: “Proposed initiatives to facilitate organicity investigations’ implementation in nonacademic departments”. 

• The conclusions in this work are based 0n personal opinions of 16 youth psychiatrists, who may be rather naïve or inexperienced to make rational recommendations. To make the conclusions stronger psychiatrists with more years of experiences need to be included in this study. Additionally, evidence from research studies, such as CRT, in support to the conclusions would be helpful.

• We thank the reviewer for this comment. We believe our answers to the previous questions and subsequent Manuscript modifications address these issues.

We agree with the reviewer regarding the usefulness of CRT in preventing relapse and improving cognitive efficacy in patients’ daily lives. (Mueller, D., & Roder, V. (2017). 101. Does Cognitive Remediation Therapy Prevent Relapses in Stabilized Schizophrenia Outpatients? A 1-Year RCT Follow-Up Study. Schizophrenia Bulletin, 43(Suppl 1), S53–S54. https://doi.org/10.1093/schbul/sbx021.139 ; Amado, I., Moualla, M., Jouve, J., Brénugat-Herné, L., Attali, D., Willard, D., ... & Yannick, M. (2020). Employment, Studies and Feelings: Two to Nine Years After a Personalized Program of Cognitive Remediation in Psychiatric Patients. Frontiers in Psychiatry, 11, 609 ; O’Reilly, K., Donohoe, G., O’Sullivan, D. et al. A randomized controlled trial of cognitive remediation for a national cohort of forensic patients with schizophrenia or schizoaffective disorder. BMC Psychiatry 19, 27 (2019). https://doi.org/10.1186/s12888-019-2018-6). However, as stated earlier, the aim of our study wasn’t to provide guidelines to improve the care of patients with psychosis, and therefore, we have not discussed the efficacy of psychological or psychosocial interventions. 

Response to Reviewer 3 : 

• page 3, last sentence. replace semicolons by commas:

• We agree with the reviewer and have modified the manuscript as follows: “They focused on their daily use of organicity investigations in their workplaces, their training and professional experience, the reactions of their colleagues, patients, and families to these organicity investigations, and their contacts with expert centers.”

• Page 4: first sentence: Sixteen "youth" psychiatrists aged 28 to 60. I suggest to remove the word "youth" given the fact that the age range goes far beyong youth.

• We thank the reviewer for this comment which has allowed us to clarify this issue. I refer you to the answer made to Reviewer 2 on this subject. 

• Page 7: from third line: very long sentence. Try to improve wording by separating sentences.

• We agree with the reviewer and have modified the manuscript as follows: “There are therefore two relational drawbacks identified by the participants. The first was the fact that the avoidance of dealing with families’ worries about their child's long-term mental health, future schooling and social integration was only slightly delayed. Secondly, they sometimes found that trying to convince their patient at any cost generated tension.”

• Page 8: "The practices of some participants remained modeled on that of the expert center they had [been] trained at." The verb on the sentence was missing.

• We agree with the reviewer and have modified the manuscript as follows: “The practices of some participants remained modeled on that of the expert center they had been trained at.”

• Page 9: wording suggestions: "Therefore, participants [were]worried [about] the

stigmatization of psychiatric patients would lead to less conscientious assessments"

• We agree with the reviewer and have modified the manuscript as follows: “Therefore, participants were worried about the stigmatization of psychiatric patients would lead to less conscientious assessments.”

• Figure 1: I don't fully grasp the difference between the second ("to transform") and the third ("to be transformed") categories. I suggest to choose other words to highlight the core ideas

Thank you for your continued interest of this submission. I look forward to your reply. 

Best regards, 

Jean-Luc Kurukgy and Laelia Benoit

---

## [Decision Letter · Decision Letter 1]

19 May 2021

Implementing organicity investigations in early psychosis: Spreading expertise

PONE-D-20-37522R1

Dear Dr. Benoit,

We’re pleased to inform you that your manuscript has been judged scientifically suitable for publication and will be formally accepted for publication once it meets all outstanding technical requirements.

Kind regards,

Gerard Hutchinson, MD

Academic Editor

PLOS ONE

Additional Editor Comments (optional):

Reviewers' comments:

Reviewer's Responses to Questions

**Comments to the Author**

1. If the authors have adequately addressed your comments raised in a previous round of review and you feel that this manuscript is now acceptable for publication, you may indicate that here to bypass the “Comments to the Author” section, enter your conflict of interest statement in the “Confidential to Editor” section, and submit your "Accept" recommendation.

Reviewer #2: All comments have been addressed

Reviewer #3: All comments have been addressed

2. Is the manuscript technically sound, and do the data support the conclusions?

Reviewer #2: Yes

Reviewer #3: Yes

3. Has the statistical analysis been performed appropriately and rigorously? 

Reviewer #2: N/A

Reviewer #3: Yes

4. Have the authors made all data underlying the findings in their manuscript fully available?

Reviewer #2: Yes

Reviewer #3: Yes

5. Is the manuscript presented in an intelligible fashion and written in standard English?

Reviewer #2: Yes

Reviewer #3: Yes

6. Review Comments to the Author

Reviewer #2: The authors have addressed all of my previous comments clearly and fully. The manuscript reads well and provides a clear outline of the challenges underlying the implementation of organicity investigations when diagnosing and treating individuals affected with psychosis.

Reviewer #3: (No Response)

7. PLOS authors have the option to publish the peer review history of their article (what does this mean?). If published, this will include your full peer review and any attached files.

Reviewer #2: No

Reviewer #3: **Yes: **Paula Herrera Gomez

---

## [Editor Report · Acceptance letter]

2 Jun 2021

PONE-D-20-37522R1 

Implementing organicity investigations in early psychosis: Spreading expertise 

Dear Dr. Benoit:

I'm pleased to inform you that your manuscript has been deemed suitable for publication in PLOS ONE. Congratulations! Your manuscript is now with our production department. 

Kind regards, 

on behalf of

Dr. Gerard Hutchinson 

Academic Editor

PLOS ONE